# A Cluster-Randomised Stepped-Wedge Impact Evaluation of a Pragmatic Implementation Process for Improving the Cultural Responsiveness of Non-Aboriginal Alcohol and Other Drug Treatment Services: A Pilot Study

**DOI:** 10.3390/ijerph20054223

**Published:** 2023-02-27

**Authors:** Sara Farnbach, Alexandra Henderson, Julaine Allan, Raechel Wallace, Anthony Shakeshaft

**Affiliations:** 1National Drug and Alcohol Research Centre (NDARC), UNSW Sydney, Sydney, NSW 2052, Australia; 2Rural Health Research Institute, Charles Sturt University, Orange, NSW 2800, Australia; 3Network of Alcohol and Drug Agencies, Woolloomooloo, NSW 2011, Australia; 4Poche Centre for Urban Indigenous Health, University of Queensland, Brisbane, QLD 4072, Australia

**Keywords:** cultural responsiveness, co-design, implementation, Aboriginal and Torres Strait Islander, alcohol and other drugs, service delivery, audit

## Abstract

There is limited evidence regarding implementing organisational improvements in the cultural responsiveness of non-Aboriginal services. Using a pragmatic implementation process to promote organisational change around cultural responsiveness, we aimed to (i) identify its impact on the cultural responsiveness of participating services; (ii) identify areas with the most improvement; and (iii) present a program logic to guide cultural responsiveness. A best-evidence guideline for culturally responsive service delivery in non-Aboriginal Alcohol and other Drug (AoD) treatment services was co-designed. Services were grouped geographically and randomised to start dates using a stepped wedge design, then baseline audits were completed (operationalization of the guideline). After receiving feedback, the services attended guideline implementation workshops and selected three key action areas; they then completed follow-up audits. A two-sample Wilcoxon rank-sum (Mann–Whitney) test was used to analyse differences between baseline and follow-up audits on three key action areas and all other action areas. Improvements occurred across guideline themes, with significant increases between median baseline and follow-up audit scores on three key action areas (median increase = 2.0; Interquartile Range (IQR) = 1.0–3.0) and all other action areas (median increase = 7.5; IQR = 5.0–11.0). All services completing the implementation process had increased audit scores, reflecting improved cultural responsiveness. The implementation process appeared to be feasible for improving culturally responsive practice in AoD services and may be applicable elsewhere.

## 1. Introduction

There are substantial inequalities in health status and health care access between Aboriginal and Torres Strait Islander people (hereafter referred to as Aboriginal) and non-Aboriginal people in Australia [1], including disproportionate drug and alcohol-related morbidity and mortality [2]. Although all health services should be culturally safe, effective and welcoming to Australians from any cultural backgrounds, there is evidence that Aboriginal people receive less benefit from non-Aboriginal health services than non-Aboriginal people [3]. Ensuring that mainstream health services (that is, services that are not specifically developed for Aboriginal people) are responsive to Aboriginal peoples’ needs is a key strategy to reduce inequalities in healthcare access and enhance the quality of care provided to Aboriginal people [4,5,6]. Cultural responsiveness is an ongoing process of adapting systems, services and practice to fit with culturally diverse user preferences [7], and providing high-quality care that is culturally appropriate and safe [8]. While the importance of culturally responsive health services is well acknowledged [8], there is a lack of consensus on effective methods to develop health services that are culturally responsive [6]. 

Cultural responsiveness initiatives have been shown to improve healthcare worker cultural knowledge, awareness and sensitivity [9,10,11,12], improve patient satisfaction with providers [9,12,13] and increase access and frequency of visits by Aboriginal people [14]. However, the quality of many existing studies is low, frequently using observational study designs and interventions that provide one-off staff training, but which tend to be ineffective if not implemented as part of a systematic approach [6,15]. 

While many non-Aboriginal clinicians are individually committed to practising in a culturally responsive way, improving cultural responsiveness needs to be a whole-of-service activity that involves multiple strategies across all levels of the workforce and organisational policy, management and practices to be effective [5,6,16,17,18]. There is limited evidence, particularly in the Australian context, regarding effective systematic methods for implementing organisational-level change to improve cultural responsiveness [6]. Aiming to provide a structured method to implement best-evidence cultural responsiveness practices, the current project developed a pragmatic implementation process for facilitating organisational change in services. The first step of this process involved combining a number of recommended cultural responsiveness strategies [19] into a best-evidence guideline for improving the cultural responsiveness of non-Aboriginal AoD services [20]. The co-designed best-evidence guideline details a wide range of evidence-based strategies including: engaging management [21,22]; enhancing communication and relationships between mainstream and Aboriginal services and communities [3,23,24]; improving staff knowledge of the social and historical determinants of health [25]; and tailoring programs to suit the local community [26]. 

The core components, or themes, of the guideline were then operationalised into flexible activities that could be tailored to suit each service [27,28,29], and implemented in non-government organisation (NGO) non-Aboriginal AoD services. The implementation fidelity, barriers and facilitators to implementation, and their acceptability and feasibility, are described elsewhere [27,30]. The current study aims to identify the impact of the implementation process on the cultural responsiveness of participating services, as measured by the mean change in audit scores from baseline to three-month follow-up. Secondary aims were to identify the areas of the guideline that were most frequently selected as priority areas for change and most successfully actioned by services during the project. We also aimed to build on the services’ insights to develop a program logic to identify how the standardised core components were flexibly applied by services to support future implementation.

## 2. Materials and Methods

### 2.1. Study Design

The project was co-designed and implemented using a community-based participatory research approach [29,31] that facilitated iterative development of the best-evidence guideline and the pragmatic implementation process through collaboration between the project team (RW and JA, who have experience working in NSW AoD services), the researchers (SF, AH, AS), the Network of Alcohol and other Drugs Agencies (NADA; the peak organisation for the NGO AoD sector in NSW), the Primary Health Networks (PHNs) as the project funders and an Aboriginal Advisory Group (which included Aboriginal community members with professional and community connections to NGO AoD treatment services or government treatment services). The project was overseen by the Aboriginal Advisory Group to ensure the priorities and world views of Aboriginal experts were centralised into the guideline and the project implementation. Members of the Group were offered reimbursement for expenses arising from their involvement. Project implementation expenses were covered by the project. The impact of the project on the cultural responsiveness of participating services was evaluated using a cluster-randomised stepped-wedge design with 12 services and six clusters.

### 2.2. Participating Services

Seventeen non-Aboriginal NGO AoD treatment services from six PHN districts in New South Wales (NSW) were identified by the PHNs as being potentially willing to participate, with fifteen providing formal consent to participate (88%) (hereafter referred to as participating services). Participating services included a variety of AoD service types, including residential rehabilitation (*n* = 3), day programmes (*n* = 2), centre-based counselling and support (*n* = 3), outreach counselling and support (*n* = 4), groupwork and phone support (*n* = 1) and group or individual youth services (*n* = 2). Twelve services completed all project activities (80%; Table 1). No data related to Aboriginal clients or organisations were accessed or used in this phase of the project.

### 2.3. Cultural Responsiveness Project 

The project was delivered in these sequential phases: (i) engage stakeholders, develop co-design structures and secure approvals from ethics and participating sites; (ii) co-design the implementation process and best-evidence guideline; (iii) implement the guideline and monitor uptake. Phase 1 and the process evaluation outcomes are described in detail elsewhere [27] and the co-design, implementation and monitoring steps are described below (phases 2 and 3). Aboriginal author RW was involved in all aspects of the project and was provided training in research methods, manuscript development and presenting research findings. Non-Aboriginal members of the research team (JA, AH, SF, AS) have extensive experience of working with Aboriginal communities over multiple projects and have completed training in cultural responsiveness. RW provided cultural mentoring to non-Aboriginal researchers. Findings from the project were presented to participating services and local Aboriginal peak bodies via ongoing discussions about the project and at formal events, such as the Aboriginal Corporation Drug and Alcohol Network of NSW (ACDAN) Symposium. 

### 2.4. Co-Designed Best-Evidence Guideline for Cultural Responsiveness in Non-Aboriginal AoD Services (Phase 2)

A best-evidence guideline that describes key elements of culturally responsive service delivery in non-Aboriginal AoD treatment services was co-designed at the beginning of the project and this process is described fully in the guideline document (20) (See Appendix A or also published online at https://www.nada.org.au/resources/alcohol-and-other-drugs-treatment-guidelines-for-working-with-aboriginal-and-torres-strait-islander-people-in-a-non-aboriginal-setting/ (accessed on 1 June 2022)). Briefly, the guideline co-design process was facilitated by an Aboriginal project team member (RW) and overseen by the Aboriginal Project Advisory Group (27). The guideline identifies six themes: (1) Creating a welcoming environment, (2) Service delivery, (3) Engagement with Aboriginal organisations and workers, (4) Voice of the community, (5) Capable staff, and (6) Organisation’s responsibilities.

### 2.5. Implementation and Monitoring Process (Phase 3)

#### 2.5.1. Clustering of Participating Services and Randomisation to a Starting Date

Services were clustered based on PHN district/geographical region (*n* = 6). Each cluster of services was randomised to an implementation starting date between June and October 2019, with approximately one month between clusters, as shown in Table 2. Cluster randomisation was conducted by a statistician independent of the project using random number generation. Owing to varying numbers of services within regions, and attrition, clusters included different numbers of services; cluster 1 included one service, cluster 5 included three services and the remaining clusters included two services each.

The following implementation and monitoring steps were completed with each participating service.

#### 2.5.2. Baseline Audits of Participating Services

Services were advised of their allocated start date and structured baseline audits of current culturally responsive practice, using a standardised audit tool, were completed individually with each participating service. The audit process identified the extent to which services addressed the guideline, rating cultural responsiveness according to 21 actions areas which corresponded with the six guideline themes. Audit tools were developed which framed the 21 action areas as questions in order to collect information from staff at participating services. Audits were conducted by two trained auditors (RW, JA or another trained auditor) in the setting where the service is delivered and took between 90 min to two hours to complete. Auditors were independent of the service being audited and at least one auditor at each audit was Aboriginal. 

#### 2.5.3. Audit Feedback to Participating Services

Individualised written feedback from the audit findings was provided to each participating service, listing all guideline action areas with a descriptive assessment for each area reflecting the level of evidence observed during the audit (limited, some, good or excellent) and recommendations for areas where potential improvements could be made. 

#### 2.5.4. Guideline Implementation Workshops with Participating Services

Implementation workshops were held with key staff from services (CEOs/managers and direct service delivery staff) to explain the guideline, review the audit feedback, set goals for improvement and develop a detailed action plan tailored to their service (to operationalise action areas from the guideline themes). Workshops were facilitated by JA and RW. Staff identified and prioritised specific activities that they would implement from the 21 action areas and were encouraged to select three key action areas for their service to progress over the next three months. For example, activities that operationalise guideline Theme 1: Creating a welcoming environment, might include processes to ensure that all clients are welcomed respectfully at first contact with the service, providing tea/coffee/water in the waiting room, accommodating children or other family members in the service, or displaying local Aboriginal artwork. These self-designed activities provide flexibility in how individual services operationalised and implemented the core components, enabling the practice change activities to be tailored to the needs and resources of individual services and the communities they serve [1,2,3,4,5]. 

#### 2.5.5. Follow-Up Audits of Participating Services 

Follow-up audits of services were conducted after three months to assess change in culturally responsive practices in the 21 action areas, following the same procedure as for the baseline audits. Where possible, the same service staff attended the follow-up audit. Services were provided with a second individualised feedback report, including discussion of any changes that had occurred.

### 2.6. Measures

To privilege Aboriginal values and views throughout analysis and reporting, we used the guideline themes that were developed by the Aboriginal Advisory Group to assess culturally responsive practices. The study aimed to identify the impact of the project on the cultural responsiveness of services using the following outcomes:Change in audit score from baseline to follow-up audit on the three key action areas identified by staff at the implementation workshops (possible score 0–9).Change in audit score from baseline to follow-up audit in all other action areas from the guideline (other than the three key action areas selected by each service (possible score 0–54).

### 2.7. Statistical Analysis

The audit responses provided by staff were recorded into the audit tool. After each audit was completed, ratings of 0–3 were allocated to each of the 21 audit criteria, according to pre-specified rating rules, by one of the researchers conducting the audit (RW). A second researcher (SF) then independently reviewed the audit tool and rated the 21 criteria. The two sets of ratings were compared and any disagreement around ratings were resolved by discussion, until a consensus was reached. A two-sample Wilcoxon rank-sum (Mann–Whitney) test was used to analyse the difference in audit scores between baseline and follow-up audits, on the three key action areas (outcome 1) and all other action areas (outcome 2). All analyses were conducted using Stata 16 [32].

The extent of change across the six guideline themes was identified by summing item ratings within each theme and calculating the rates of change for each theme. The frequency with which each individual action area was selected by service staff (during workshops to operationalise their improvement goals), and whether improvements were subsequently observed in those action areas, were descriptively explored.

### 2.8. Development of a Program Logic

We used a program logic structure developed in previous work [5,7] to build a standardised logic model specifically for improving cultural responsiveness in non-Aboriginal NGO AoD services. The program logic model was developed by reviewing the audit findings and activities chosen by staff during the workshop and linking these to the core components (themes) of the guideline. 

## 3. Results

### 3.1. Implementation Process

Twelve of the fifteen participating services completed all service-specific project components. Some delays in completing the three-month follow-up audits occurred, with an average time between audits of 18 weeks (range 14–28 weeks) (See Table 2). The longest delays in completing the follow-up audits were for services “J”, “D” and “F”, with the audits completed at 19, 24 and 28 weeks, respectively. Service “B” was part of cluster 5; however, due to delays in completing the baseline audit, service “B” ultimately completed the project components in line with cluster 6. Further detail on implementation and process outcomes are reported elsewhere [27].

### 3.2. Change in Cultural Responsiveness of Services in Three Key Action Areas 

Outcomes are reported for services that completed baseline and follow-up audits (*n* = 12). 

Ten of 12 services increased their audit score on their three key action areas at follow-up. The median follow-up scores were statistically significantly higher than the median baseline scores (median change = 2.0, IQR = 1.0–3.0, z = −2.79, *p* < 0.005) (Table 3).

### 3.3. Change in Cultural Responsiveness of Services in All Other Action Areas 

All 12 services showed an increase in score on all other action areas (excluding the three key action areas). The median follow-up scores were statistically significantly higher than the median baseline scores (median change = 7.5, IQR = 5.0–11.0, z = −1.97, *p* < 0.05) (Table 3). 

### 3.4. Guideline Themes with the Most Improvement 

Overall, there were improvements in scores across all six themes of the guideline and all showed similar rates of improvement; Theme 5: Capable staff (+22%), Theme 3: Voice of the community (+18%), Theme 6: Organisation’s responsibilities (+18%), Theme 1: Creating a welcoming environment (+17%), Theme 2: Service delivery (+16%) and Theme 4: Engagement with Aboriginal organisations and workers (+16%). 

### 3.5. Action Areas Most Frequently Selected (by Staff) and Most Frequently Improved 

Service staff chose a wide variety of action areas from the guideline to prioritise; 16 of the 21 areas were selected at least once. Those most frequently selected as key action areas were: 1B: The physical environment is welcoming to Aboriginal people (*n* = 6 services); 3Ai: Aboriginal community engagement to develop relationships (*n* = 4); 3Aiii: Local history and protocols are reflected in practice and/or policy (*n* = 5); and 4A: Developing connections with Aboriginal organisations and workers (*n* = 5) (see the guidelines in Appendix A for further description). The action areas that services most frequently improved on were in Theme 2 (2B: Immediate triage options are available for Aboriginal people (*n* = 8 services) and 2C: Staff are culturally responsive in therapeutic practice (*n* = 7)), Theme 3 (3B: Local Aboriginal protocols are reflected in practice and/or policy (*n* = 7)) and Theme 6 (6Aii: There are Aboriginal-identified positions and Aboriginal publications and networks are used to advertise jobs (*n* = 7), 6Aiii: Service induction includes materials about working with Aboriginal people and materials are developed/reviewed by a local Aboriginal person (*n* = 8)).

### 3.6. Development of a Program Logic

To facilitate future implementation of improvements to cultural competence, the researchers (AH, AS, SF) developed a program logic that is directly tied to guidelines, shown in Table 4. This program logic was developed post-implementation to clearly delineate the standardised core components (guideline themes), flexible components (service level activities) approach and likely mechanisms of change, for future iterations of this project, based on previous work by the authors [28,29]. The second column lists the six best-evidence themes/principles that comprise the core components of the guideline. These are standardised across all services, as are the aims/goals/target areas for improvement (first column) and the articulation of why these core components would impact on cultural responsiveness (third column). The fourth column provides examples of specific activities that services can implement, with flexibility to choose practice change activities tailored to the needs and resources of individual services [28,31,33,34,35]. The remaining columns identify the measures of processes (the extent to which services engaged in the intervention process), outcomes (the extent to which indicators of culturally responsive practice improved) and data sources.

## 4. Discussion

The current study used a community-based, participatory research approach to develop a best-evidence, service-level practice change process, supported by a program-logic framework previously developed by the authors [28,29]. The pragmatic implementation approach is supported by existing evidence [33,34,35] and means that individual services could implement areas of the guideline that were most relevant to their local context and current level of cultural responsiveness. All participating services increased their overall audit scores and most increased scores on their chosen priority action areas, reflecting an increase in compliance with the guidelines and improved cultural responsiveness. The results are consistent with previous research demonstrating that audits and practice improvement interventions can be effective methods of identifying where improvements are needed, engaging with workers, and improving culturally responsive practices in a variety of health settings [14,36,37,38]. Our results support the effectiveness of the guideline and implementation process as a meaningful way of identifying and operationalising best-evidence principles of cultural responsiveness and enabling staff to understand and enact components of the guidelines that were relevant to their service. The program-logic model links the best-evidence core components of the guideline to the flexible service level activities, likely mechanisms of change, processes and outcomes, and can be used to guide future work on improving cultural responsiveness in AoD and potentially other health and human services. Our approach demonstrates a process to improve the cultural responsiveness of service delivery, and it is hoped that this approach impacts on inequalities in health status and healthcare access between Aboriginal and non-Aboriginal people.

An important strength of the project is that the co-design and implementation was led by an Aboriginal researcher and AoD worker, with extensive consultation with senior Aboriginal AoD clinicians (via the Aboriginal Advisory Group), funders, researchers, as well as links with workers (via the peak organisation for the NGO AoD sector) [6,21,24]. Furthermore, the guidelines recommend multiple evidence-based strategies across all organisational levels [6,16], such as: tailoring of service delivery to local communities [26]; enhancing relationships with Aboriginal services and communities [3,23,24]; improving staff knowledge and competency [25]; and implementing organisation-wide policies and practices [39]. The improvements in audit scores observed across all themes of the guideline, indicating that these concepts and strategies were clearly operationalised. Improvements were frequently observed in areas related to enhancing relationships with Aboriginal communities (e.g., having local Aboriginal protocols reflected in practice and/or policy), improving staff knowledge and skills (e.g., improved crisis triage options and staff demonstrating cultural responsiveness in direct service delivery) and organisation-wide policies or practices (e.g., including materials about working with Aboriginal people in service induction training). A larger evaluation with a longer timeframe would allow a more detailed exploration of specific components of the audits and guidelines and whether there are critical activities that services can enact to improve cultural responsiveness.

In addition to measuring change in audit scores (reflecting change in cultural responsiveness), future studies should also aim to examine the impact of these changes on service delivery or utilisation outcomes, potentially through using routinely collected administrative data. Previous reviews of cultural responsiveness programs have highlighted the need for valid indicators of change and objective outcome measures [6,12], and routinely collected administrative data represent objective, pragmatic, low-cost and easily tracked outcomes. As services improve their levels of cultural responsiveness, we would hope to also see improvements in service utilisation by Aboriginal people (for example, the number of episodes of care provided to and completed by Aboriginal people). The short time frame of the current evaluation limited our ability to examine these types of outcomes. Not only was the time for services to enact changes limited [27], but three months was likely not sufficient for any changes implemented to impact on service utilisation or client outcomes.

The project used a methodologically strong assessment process involving a standardised audit tool that reflected the best-evidence guidelines and a double-scoring system to enhance inter-rater reliability [40]. The possibility of practice effects should be noted; service staff may have had a more thorough knowledge of the audit criteria after completing the baseline audit, leading to more positive reporting of activities in follow-up audits. The practical implication of this is that some of the improvement in follow-up audit scores may be due to improvements in staff understanding of the audit, rather than the specific cultural competence activities they enacted. This is an issue about the true mechanisms of change: it is likely that the observed changes in cultural responsiveness are a combination of both the activities themselves and greater familiarisation with the audit process and content. Some services had limited capacity for improvement in audit scores for their three priority action areas; three services chose to prioritise an area that already had a full score at baseline, and one service only selected two priority areas. For future implementations, services should be encouraged to choose priority areas that have room for improved practice, providing maximum opportunity for improvements.

Participating services were self-selected, and it is possible that they may have had a pre-existing active interest in and/or resources to dedicate to improving their cultural responsiveness. The significant improvements in audit scores achieved by these services may not occur so quickly in other services. However, the participating services do represent a broad geographic and demographic area of NSW (including both urban and regional locations), as well as a variety of service delivery types. Service frontline and managerial staff rated the project as highly acceptable [27]. A key next step is a longer-term follow-up of participating services to establish whether the improvements in culturally responsive practice can be maintained or extended over time. Importantly, in line with the logic model presented, this will include an examination of administrative data to assess any changes in service utilisation. Then, if indicated, a randomised controlled trial evaluation of the implementation process in a larger sample of services may be warranted to demonstrate the generalisability, and costs and benefits of the process.

## 5. Conclusions

The co-designed best-evidence guideline and pragmatic implementation process represents a feasible and acceptable method (27) for implementing service-wide improvements in cultural responsiveness and may be applicable to improving the cultural responsiveness of a wide variety of health and human services. The randomised stepped-wedge evaluation design, double-rated audit scoring, and standardised core intervention increased methodological rigour, while the flexibility with which individual services can operationalise and implement the guidelines allowed tailoring to available resources and needs.

## Figures and Tables

**Table 1 ijerph-20-04223-t001:** Study recruitment and retention.

*n* = 17 services invited to participate		
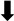		
*n* = 15 services consented to participate (88%)	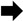	*n* = 2 services did not provide consent
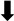		
*n* = 14 services completed baseline audit (93%)	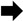	*n* = 1 service did not complete due to limited staff/time available
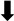		
*n* = 13 services had staff attend workshop and completed action plan (87%)	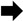	*n* = 1 service did not complete due to staff turnover and non-response
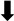		
*n* = 12 services completed 3-month audit (80%)	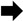	*n* = 1 service did not complete due to limited staff/time available

**Table 2 ijerph-20-04223-t002:** Stepped-wedge progression of clusters from wait-list control to the baseline audit, workshop and follow-up audit.

	Year	2019								2020			
	Month	May	Jun	Jul	Aug	Sep	Oct	Nov	Dec	Jan	Feb	Mar	No. of Weeks between Audits
Cluster	Service Code												
1	K		11/06 ^a^ 20/06 ^b^			3/10 ^c^							16
2	L			17/07 31/07			30/10						15
	C			24/07 31/07			31/10						14
3	F				09/08 16/08					09/02			28
	J				08/08 16/08			3/12					19
4	H					28/08 10/09			16/12				16
	I					29/08 10/09			18/12				16
5	D						24/09 09/10				10/03		24
	E						25/09 09/10			30/01			18
	B							25/10 28/11			28/02		14
6	A							23/10 07/11			28/02		17
	G							23/10 07/11			28/02		17

^a^ Date of baseline audit; ^b^ date of implementation workshop; ^c^ date of follow-up audit. Grey section indicates intervention period.

**Table 3 ijerph-20-04223-t003:** Change in audit scores for the three key action areas and all other action areas.

Cluster	Service	Baseline Score on 3 Key Areas (Range 0–9)	Follow-Up Score on 3 Key Areas (Range 0–9)	Change in Score on 3 Key Areas	Baseline Score on All Other Areas (Range 0–54)	Follow-up Score on All Other Areas (Range 0–54)	Change in Score on All Other Areas
1	K	2	7	+5	11	29	+18
2	C	4	7	+3	28	40	+12
	L	4	4	0	37	46	+9
3	F	3	8	+5	30	41	+11
	J	2	4	+2	13	24	+11
4	H	5	8	+3	22	27	+5
	I	2	3	+1	22	25	+3
5	B	4	5	+1	33	36	+3
	D	3	3	0	7	13	+6
	E	3	6	+3	31	36	+5
6	A	5	7	+2	26	31	+5
	G	2	4	+2	28	39	+11
Median (IQR)		3.0 (2.0–4.0)	6.0 (4.0–7.0)	+2.0 * (1.0–3.0)	27.0 (19.8–30.3)	34.0 (26.5–39.3)	+7.5 * (5.0–11.0)

* *p* < 0.05; IQR = Inter Quartile Range.

**Table 4 ijerph-20-04223-t004:** Program logic model for cultural responsiveness practice improvement.

Goals or Target Problem		Program to Be Delivered		Measures		
	Best-Evidence Core Components (Themes/Action Areas)	Why Would This Component Work?	Examples of Flexible Activities That Can Be Implemented By Services ^a^	Process	Outcomes	Data Source for Process/Outcomes
**Target problem** Some Aboriginal Alcohol and other Drug (AoD) clients will access non-Aboriginal AoD services. There is a lack of guidance for non-Aboriginal services around processes involved with culturally competent service delivery. **Goals** Use a co-designed best-evidence guideline for culturally responsiveness practice to assist services in improving cultural responsiveness. Services will be able to achieve changes (improvements) to cultural responsiveness through a range of one-off or ongoing activities that relate to Guideline Themes	* **1: Creating a Welcoming Environment** * **A.** Welcoming greeting **B.** Physical environment	***A***-A welcoming first impression will keep people coming back. When a service acknowledges Aboriginal people, it demonstrates cultural respect. ***B***-Improved access for clients by accommodating for children or other family members to attend the service with clients.	Ensure clients are greeted respectfully by a staff member at first contact with the service Have a process/space to accommodate families or children Display Aboriginal artwork/ posters/ materials in client-facing areas	These track whether project components and activities are being implemented: i. Fidelity of implementation of each step of the project in each service ii. Acceptability of the program to CEOs/ managers, and direct service staff iii. Enablers and barriers to implementation	These track whether the activities impact on cultural responsiveness: i. Increase in audit rating in three key areas and all other areas ii. Changes in service utilisation ^b^ iii. Changes to clients’ experiences or outcomes ^b^	i. Implementation and evaluation log ii. Staff survey and interviews with CEOs/managers iii. Staff survey and interviews with CEOs/managers iv. Audit outcomes (pre versus post) v. Administrative service use data ^b^ vi. Patient-reported experience measures (PREMS)/ Patient-reported outcome measures (PROMS) ^b^
	* **2: Providing flexible and culturally informed service delivery options** * **A.** Service delivery **B.** Referrals and assessments **C.** Direct practice	***A***-Providing flexible service delivery options may increase visibility and access to a service. Providing services via outreach can create opportunities for access by people who might otherwise not attend. ***B***-Seeing the same staff member each time can establish trust and rapport. Yarning to someone before requiring them to fill out forms or answer personal questions helps build rapport. ***C***-Direct practice strategies can engage Aboriginal people and their families.	Ensure clients can access services or staff via outreach, at Aboriginal services or home visits Clinical outcome measures and/or assessment tools are developed for use by Aboriginal people Establish crisis/quick triage referrals that meet the needs of Aboriginal people Employ direct practices that are specifically developed to be inclusive of Aboriginal people, e.g., strengths-based and narrative approaches			
	* **3: Having the voice of the Community within the service** * **A.** Community engagement **B.** Local history and protocols	***A***-When a service is involved in the community it may help to build relationships, trust and awareness of the service. By building relationships within the community, the service may become more known from people talking or ‘vouching’ for it within community. ***B***-Having knowledge of local history, culture and protocols is critical to understanding the local Aboriginal community. Recognition of local language groups demonstrates respect.	Attend events such as Sorry Day, NAIDOC, football games and sports days repeatedly. Create a plan to introduce staff to different services and groups to build relationships. Consumer groups with Aboriginal representation Identify localised training offered by Elders or Local Aboriginal Land Councils Contact the Local Aboriginal Land Council for more details around traditional custodianship			
	* **4: Engagement with Aboriginal Organisations and Staff** * **A.** Organisation and staff **B.** Projects **C.** New services/programs establishment	***A***-Aboriginal organisations may not provide all types of services and most will be interested in working with mainstream services to fill gaps. ***B***-Collaborative projects, shared care and developing referral pathways to and from Aboriginal services are good ways to share resources with those services. ***C***-New services/programs for Aboriginal people should be developed in consultation with the Aboriginal Community Controlled Organisation (ACCHO) and community.	Identify service gaps Create a local service directory of Aboriginal organisations and services. Identify ways of working together or propose shared projects, particularly with local ACCHO. With good partnerships, consider shared service delivery for new projects or programs. Work closely with the ACCHO by consulting prior to the development of any program or project.			
	* **5: Capable Staff** * **A.** Staff knowledge and skill assessment **B.** Clinical/practice supervision	***A***-Staff should know about Australia’s history of dispossession and its impact on Aboriginal people, transgenerational trauma, the impacts and how to support clients impacted by transgenerational trauma. Education on men’s and women’s business, including sensitive topics, is critical for good communication and avoiding embarrassment for clients. ***B***-Clinical or practice supervision is an important way to develop and practice skills and address challenges. Supervision should have a cultural lens with an appropriate supervisor who is selected for their practice experience with Aboriginal people.	Provide training and support in working with transgenerational trauma. Provide training and advice about gender roles and communication for staff. Assess the impact of training by asking for descriptions of or observing practice change. A process to support and debrief staff when they are involved in critical incidents relating to Aboriginal service users. Cultural supervision with a well-respected local Aboriginal community member.			
	** *6: Organisational Responsibilities* ** **A.** Employment practices **B.** Aboriginal staff support **C.** Training provided **D.** Organisation wide practices **E.** Policies and procedures	***A***-Recruitment of Aboriginal staff should be through Aboriginal networks. It is important to have Aboriginal people on interview panels to hear their perspective about interviewees’ performance. Providing new staff with materials about the local Aboriginal history and culturally appropriate communication shows that these skills are valued. ***B***-Aboriginal AoD workers face many unique stressors including family and community responsibilities. Strategies to support and retain Aboriginal staff and prevent burnout are required. ***C***-Training related to working with Aboriginal people, families and communities should be actively promoted to staff, embedded into learning and development plans and made compulsory. Training should include all staff, not just service delivery staff, and where possible be specific to their role. ***D***-Reconciliation Action Plans (RAP) enable organisations to contribute to reconciliation by building and encouraging relationships between Aboriginal peoples, communities, organisations, and the broader Australian community. Representation on an organisation’s Board is an important way to involve Aboriginal people in governance and the overall approach to Aboriginal people within the service. ***E***-Policy and procedure support all staff to follow the same processes, ensuring management support for practice methods and creating consistency over time as staff change. Some policies and procedures may need to include how the service will work with and relate to Aboriginal people, e.g., community engagement policy, an intake policy that includes asking about Aboriginal status or whether there is a preferred clinician.	Identify and evaluate the history of Aboriginal people working in your organisation including recruitment strategies and retention. Have Aboriginal people on interview panels Employ Aboriginal staff in mainstream positions, not just in identified roles, as well as creating identified roles Use the NCETA Feeling Deadly, Working Deadly resource kit. Actively plan retention and career paths for Aboriginal people. Identify compulsory cultural competence training for all staff. Identify local training for all staff and repeat it regularly. Recruit Aboriginal Board members Review complaints procedures Plan for and allocate resources to developing a RAP Review policies and procedures identified in these Guidelines to ensure compliance Ensure all staff know the policy and can apply the procedures; provide training			

^a^ The flexible activities are the actual activities that each service would need to select to operationalise (or activate) each best-evidence response component. Selected by each service. Activities listed here are illustrative examples only. ^b^ Outcomes have not yet been completed in this project, so the data are not presented in the current paper.

## Data Availability

The datasets used and/or analysed during the current study are available from the corresponding author on reasonable request. The data are not publicly available due to potential for re-identification of participating services.

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
