# Peer review of "A Cluster-Randomised Stepped-Wedge Impact Evaluation of a Pragmatic Implementation Process for Improving the Cultural Responsiveness of Non-Aboriginal Alcohol and Other Drug Treatment Services: A Pilot Study"

_ijerph, 2023, doi:10.3390/ijerph20054223_

Round 1

Reviewer 1 Report

I have reviewed the manuscript entitled “A Cluster Randomised Stepped-Wedge Impact Evaluation of a Pragmatic Implementation Process for Improving the Cultural Responsiveness of Non-Aboriginal Alcohol and Other Drug Treatment Services: A Pilot Study”. The theme developed is very interesting and original. It allows to instigate other researchers using audit methodology to improve best practices in healthcare. I believe that the readers of the journal, especially those with an interest in cultural responsiveness and inequalities in healthcare, will appreciate your work as I have.

I have nothing to add and I wish you all the best with its publication.

Best regards 

The reviewer.

Author Response

Thank you for this comment.

Reviewer 2 Report

Very interesting study, well-written paper, and recommended to be accepted in the oresent form.

Author Response

Thank you for this comment.

Reviewer 3 Report

Thank you for the opportunity to review this paper. The missing part of this paper is the explanation of stepped-wedge design apart from using independent statistician and the implication in statistical analysis. Nonparametric descriptive statistics is more appropriate than parametric statistics. Since the stepped-wedge design is used, the author should analyse the impact of implementation time in addition to the cluster.

Author Response

Thank you for considering our methods. Although our data were approximately normal, we have reviewed our stats plan and agree that using a nonparametric method is slightly more appropriate. Therefore, we have updated the test using nonparametric equivalent (Mann-Whitney Wilcoxon). We have updated results in the resubmitted version of the paper. (Results are very similar to parametric test.)

Regarding the impact of time, yes we agree that there is a time element but we did not actually have repeated measurement on each cluster which would have allowed us to estimate secular trends. Consequently, we analysed and presented all the data that we have available.

Reviewer 4 Report

- authors should consider and discuss the role of care manager in such a context. They can consider the paper from Ciccone MM et al. Vasc Health Risk Manag. 2010 May 6;6:297-305.

- please include comparisons with literature in the discussion section

Author Response

Thank you for considering this work with regard to other literature and providing this citation. We have considered this paper about care managers alongside our findings and although it is related, we believe it is not directly relevant to the points we wanted to discuss arising from out study, so have opted to keep the discussion in its current format.

Reviewer 5 Report

This work is interesting. I accept it in the present format. 

Author Response

Thank you for this comment.

Round 2

Reviewer 3 Report

The authors addressed the comments that I had proposed. However, the abstract should also be revised in addition to the main content.

Author Response

Thank you. We have included the revised stats methods into the abstract.
